# Genome-Wide Linkage Mapping of Root System Architecture-Related Traits Under Drought Stress in Common Wheat (*Triticum aestivum* L.)

**DOI:** 10.3390/plants14193023

**Published:** 2025-09-30

**Authors:** Yirong Jin, Guiju Chen, Xiaodong Qiu, Fuyan Wang, Hui Jin, Liang Zhang, Cheng Liu, Jianjun Liu, Wenjing Li, Peng Liu

**Affiliations:** 1Dezhou Academy of Agricultural Sciences, Dezhou 253015, China; jyr2014@163.com (Y.J.);; 2Wheat Research Institute, Jining Academy of Agricultural Sciences, Jining 272000, China; 3Department of Science and Technology of Shandong Province, Jinan 250101, China; 4Institute of Forage and Grassland Sciences, Heilongjiang Academy of Agricultural Sciences, Harbin 150086, China; 5Crop Research Institute, Shandong Academy of Agricultural Sciences, Jinan 250100, China; 6College of Life Science, Langfang Normal University, Langfang 065000, China

**Keywords:** bread wheat, drought tolerance, marker-assisted selection, root system architecture (RSA)

## Abstract

Drought severely threatens wheat production. Under drought conditions, root system architecture (DRSA)-related traits in common wheat significantly affect wheat production. In China, Zhoumai16 is a high-yield winter wheat variety in the Huang-Huai wheat region. It is suitable for high-fertilizer and high-water cultivation and has moderate drought tolerance. DK171 is a newly developed high-yield and stress-tolerant variety, with higher drought tolerance. Thus, identifying genetic loci associated with DRSA-related traits from DK171 and developing available molecular markers are of great importance for enhancing wheat stress tolerance breeding. In this study, DRSA-related traits, including the total root dry weight (DDRW), total root length (DTRL), total root area (DTRA), and the number of root tips (DNRT) under drought stress, were assessed using the hydroponic system in Zhoumai16/DK171 recombinant inbred lines (RIL) population. A total of five quantitative trait loci (QTL) for DRSA-related traits were identified, e.g., *QDDRW.daas-1BL*, *QDTRS.daas-4AL*, *QDNRT.daas-4DS*, *QDTRL.daas-3AL*, and *QDDRW.daas-5D*, and explained 6.1% to 18.9% of the phenotypic variances, respectively. Among these, *QDTRS.daas-4AL* and *QDTRL.daas-3AL* were consistent with previous reports, whereas the *QDDRW.daas-1BL*, *QDNRT.daas-4DS*, and *QDDRW.daas-5D* are novel. The favorable alleles of *QDTRS.daas-4AL* and *QDNRT.daas-4DS* were inherited from Zhoumai16, whereas the favorable alleles for *QDDRW.daas-1BL*, *QDTRL.daas-3AL*, and *QDDRW.daas-5D* were contributed by DK171. Furthermore, five kompetitive allele-specific PCR (KASP) markers, *Kasp_1BL_DTRS* (*QDDRW.daas-1BL*), *Kasp_3AL_DTRS* (*QDTRL.daas-3AL*), *Kasp_4A_DTRS* (*QDTRA.daas-4A*), *Kasp_5D_DDRW* (*QDDRW.daas-5D*), and *Kasp_4D_DNRT* (*QDNRT.daas-4D*), were developed and validated in a diverse panel with 108 wheat varieties mainly from China. Additionally, eight candidate genes related to plant hormone regulation, ABC transporters, and calcium-dependent lipid-binding domain proteins were identified. This study offers new loci, candidate genes, and available KASP markers for wheat drought tolerance breeding and facilitating progress in developing drought-tolerant wheat cultivars.

## 1. Introduction

Common wheat is one of the most crucial staple crops for human consumption and serves as the primary food crop in arid and semi-arid regions worldwide. Drought is the most significant abiotic stress affecting wheat production, severely constraining its sustainable cultivation [1]. Agricultural water inputs are required, resulting in considerable waste of labor and water resources. Therefore, breeding drought-tolerant and water-efficient varieties is the most effective, economical, and environmentally safe approach to reduce the losses caused by drought [2,3,4].

Wheat root system morphology and physiological regulation determine its drought tolerance. Drought tolerance of wheat seedlings is crucial for plant development and production. Root length, surface area, and number of root tips are key root system architecture under drought conditions (DRSA) traits [5,6,7,8]. These characteristics determine the root spatial distribution and primary composition, significantly affecting water and nutrient uptake. Maximum root length forms the basis for water uptake, while the root tip number, root diameter, root length, and root density are closely related to root dry weight, volume, and surface area. Together, these traits determine the plant anchoring strength in the soil profile and its capacity to absorb soil solutions [9,10]. However, the evaluation of these traits is limited by environmental conditions and measurement methods, making it time-consuming and labor-intensive [7,9,10]. Therefore, identifying stable and effective loci for wheat drought tolerance, developing molecular markers for breeding, and elucidating the genetic mechanisms of DRSA-related traits will accelerate the efficient improvement of drought tolerance.

Recently, advancements in high-throughput genotyping technologies, such as re-sequencing and SNP chips [11], effective analysis tools, such as linkage mapping [12,13,14,15,16], and association mapping [17,18] have become common methods for identifying loci of complex traits [6,8,12,13,14]. Over the past two decades, over 50 loci associated with DRSA-related traits have been reported and mainly distributed on chromosomes 1A, 2B, 3A, 3B, 5B, and 6D [6,8,19,20,21]. Although some genetic loci related to root traits in wheat seedlings under drought stress have been reported, many of these loci are linked to SSR markers, which are not efficient for practical use. Additionally, some loci are influenced by complex genetic backgrounds or are linked to non-desirable agronomic traits, making them unsuitable for breeding applications. Therefore, discovering new genetic loci and developing molecular markers that are applicable in breeding holds significant importance for optimizing wheat root systems and achieving high and stable yields. Zhoumai16 (Zhou8425B/Zhou 9) is a high-yield, disease-resistant wheat variety with a significant cultivation area in the Huang-Huai wheat region of China. It is suitable for high-fertilizer and high-water cultivation and has moderate drought tolerance. DK171 (Liangxing66/Shixin828) is a newly developed high-yield and stress-tolerant variety in recent years, with strong drought resistance inherited by Shixin828. In this study, five loci for seedling-stage DRSA-related traits were identified in the Zhoumai16/DK171 recombinant inbred line (RIL) population using the wheat 90K SNP array. The main goal of this study is to uncover the genetic basis of DRSA traits and develop breeding-friendly kompetitive allele-specific PCR (KASP) markers to enhance wheat DRSA-related trait improvement.

## 2. Results

### 2.1. Phenotypic Evaluation

Zhoumai16 is a high-yielding variety suitable for high-fertility, water-rich regions, while DK171 is a wheat variety that combines high yield with water conservation. For Zhoumai16, the means of total root length (DTRL), total root surface (DTRS), number of root tips (DNRT), and dry root weight (DDRW) were 35.9 cm, 8.0 cm^2^, 123.6, and 0.0312 g, respectively. For DK171, these values were 47.2 cm, 8.9 cm^2^, 168.9, and 0.0339 g. The DTRL, DTRS, DNRT, and DDRW of DK171 were significantly higher than Zhoumai16 (*p* < 0.05). All four DRSA-related traits exhibited continuous and significantly wide variation across the 262 RILs (Table A1 and Figure 1 and Figure A1). The means of DTRL, DTRS, DNRT, and DDRW were 40.2 cm (range: 24.1–63.8 cm), 7.8 cm^2^ (range: 5.4–10.7 cm^2^), 106.9 (range: 24.7–213.7), and 0.0282 g (range: 0.0162–0.0413 g). The standard deviation and coefficient of variation for DTRL, DTRS, DNRT, and DDRW were 6.58 cm (16.4%), 1.06 cm^2^ (13.6%), 35.7 (33.4%), and 0.0041 g (14.5%). Significant correlation was observed between DDRW, DTRL, DTRA, and DNRT, with a correlation coefficient of 0.603 (*p* < 0.05) between DTRL and DTRS (*R*^2^ = 0.56), DTRL and DNRT (*R*^2^ = 0.31), DTRL and DRW (*R*^2^ = 0.32), DTRS and DNRT (*R*^2^ = 0.33), and DNRT and DRW (*R*^2^ = 0.30) (*p* < 0.05).

### 2.2. QTL Identification

This genetic map includes all 21 chromosomes, with red representing QTL and black lines representing backbone SNP markers.

Two QTL for DDRW were identified on chromosomes 1BL and 5DL, referred to as *QDDRW.daas-1BL* (*wsnp_Ex_rep_c67299_65845319*-*Excalibur_rep_c107035_354*) and *QDDRW.daas-5D* (*BobWhite_c5176_1164-RAC875_rep_c78046_324*), respectively. These QTLs explained 18.9% (additive effect: −0.002 g) and 7.5% (additive effect: −0.0007 g) of the total phenotypic variances (PVEs) (Table 1; Figure 1). *QDTRS.daas-4AL* for DTRS was identified on 4AL chromosome (613.6–615.2 Mb, *IAAV7132-wsnp_JD_c38619_27992279*) and explained 9.6% of the PVEs with additive effect −0.275 cm^2^. *QDNRT.daas-4DS* for DNRT was identified on chromosome 4DS (1.2–3.6 Mb, *RAC875_rep_c76650_164-Kukri_c15720_884*) and explained 8.1% of the PVEs with additive effect -7.6. *QDTRL.daas-3AL* for DNRT was identified on chromosome 3AL (650.4–659.4 Mb, *Kukri_rep_c69970_717-Kukri_rep_c103783_1380*) and explained 6.1% (additive effect: −1.272 g) of the PVEs. The favorable allele of *QDTRL.daas-3AL*, *QDDRW.daas-1BL* and *QDDRW.daas-5D* were contributed by DK171, whereas the favorable allele of *QDTRS.daas-4AL* and *QDNRT.daas-4DS* were contributed by Zhoumai16 (Table 1; Figure 2).

### 2.3. Candidate Genes Identification

Eight candidate genes were identified and involved in the biological metabolism, including the plant hormones, ABC transporter and calcium-dependent lipid-binding domain protein. Two candidate genes for *QDDRW.daas-1BL* were identified, e.g., *TraesCS1B01G356600* and *TraesCS1B01G373900*, encoded the auxin-responsive protein and the ABC transporter family protein, respectively. Both *TraesCS3A01G406200* and *TraesCS3A01G416300* were candidate genes for *QDTRL.daas-3AL* and encoded the gibberellin 20 oxidase and the auxin transport protein, respectively. For *QDTRS.daas-4AL*, *TraesCS4A01G326400* were selected as the candidate gene and encoded the ethylene-responsive transcription factor. *TraesCS4D01G001900* (*QDNRT.daas-4DS*) encoded the calcium-dependent lipid-binding domain protein. *TraesCS4D01G002400* of *QDDRW.daas-5D* encoded the ethylene-responsive transcription factor. *TraesCS5D01G285900* identified at the genetic interval of *QDDRW.daas-5D* and encoded the auxin-induced in root cultures protein (Table 2 and Table A2, Figure 3). The expressions of the seven candidate genes in Zhoumai16 and DK171 were detected using the qRT-PCR. Of these, *TraesCS1B01G356600*, *TraesCS4D01G002400*, and *TraesCS5D01G285900* showed no significant differences between the parents, whereas *TraesCS4D01G002400* and *TraesCS5D01G285900* showed more than 2.3–3.5 folds higher expression in Zhoumai16 compared to DK171; *TraesCS3A01G416300*, *TraesCS4A01G326400*, and *TraesCS4D01G001900* showed more than 3.1–4.3 folds higher expression in DK171 compared to Zhoumai16 (Figure 4).

### 2.4. QTL Validation

All five QTLs were employed in the development of KASP markers. A total of 5 KASP markers, *Kasp-1BL-DDRW* (*QDDRW.daas-1BL, wsnp_Ex_rep_c67299_65845319*, 586.3 Mb) and *Kasp-3AL-DTRL* (*QDTRL.daas-3AL, Kukri_rep_c69970_717*, 650.4 Mb), *Kasp_DTRS_4A* (*QDTRL.daas-4A*, *wsnp_Ex_c7280_12498193*, 725.6 Mb), *Kasp_DDRW_5D* (*QDDRW.daas-5D*, *IACX2960*, 347.5 Mb) and *Kasp_DNRT_4D* (*QDNRT.daas-4D*, *Kukri_rep_c68594_530*, 12.7 Mb), were successfully developed. To validate the efficacy of the 5 KASP markers, a diverse panel of 108 cultivars was employed. For *Kasp-1BL-DDRW*, the allele (AA) account for 60.2% (mean DDRW: 0.0262 g) exhibited lower DDRW compared to the allele (GG), which account for 38.9% with mean DDRW 0.0230 g (*p* < 0.05). For *Kasp-3AL-DTRL*, the allele (AA) account for 66.7% (mean DTRL: 63.52 cm) exhibited higher DTRL compared to the allele (GG), which account for 32.4% with mean DTRL of 56.60 cm^2^ (*p* < 0.05). For *Kasp_DTRS_4A*, the allele (AA) account for 38.9% (mean DTRL: 5.968 cm^2^) exhibited lower DTRS compared to the allele (GG), which account for 55.6% with mean DTRS 6.993 cm^2^ (*p* < 0.05). For *Kasp_DDRW_5D*, the favorable allele (CC 10.2%, a mean DRW of 0.0297 g) showed higher DDRW than unfavorable allele (TT, 88.9%, mean DDRW of 0.0244 g) at *p* = 0.05 level. For *Kasp_DNRT_4D*, the favorable allele (CC 65.7%, mean DNRT of 97.8) showed higher DNRT than the unfavorable allele (TT, 30.6%, mean DNRT of 83.0) (*p* = 0.05) (Table 3, Table 4 and Table A3).

To assess the accuracy of markers in detecting root phenotypes in natural populations, we selected the median value as the threshold for each trait. Phenotype values below the median were categorized as non-superior phenotypes, while those above were classified as superior phenotypes. We calculated the consistency rate between genotypes and phenotypes to provide a reference for the detection and evaluation of wheat root systems under drought conditions. The accuracy rates of *Kasp_1BL_DTRS*, *Kasp_3AL_DTRS*, *Kasp_DTRS_4A*, *Kasp_DDRW_5D*, and *Kasp_DNRT_4D* in detecting superior and non-superior phenotypes for DTRS, DDRW, and DNRT were 66.0% and 75.5%; 90.6% and 92.5%; 60.4% and 73.6%; 58.5% and 79.2%; 52.8% and 66.0%, respectively.

## 3. Discussion

Roots are the main parts of plants that take up water and nutrients. They also provide support and stability. Wheat has a fibrous root system. As it grows after germination, it develops adventitious roots. These roots are vital for anchoring the plant and absorbing water and nutrients [2,13]. They form the genetic foundation for desirable traits like drought tolerance, salt tolerance, and resistance to falling over (lodging). However, studying the root system of mature plants is difficult. Soil conditions and farming practices easily affect it. Traditional methods for taking root samples are often destructive and complicated, making it hard to measure root traits efficiently [5,14]. Research shows that the root systems of young seedlings are different. They are strongly inherited and less affected by the environment. These young roots can indicate the shape and spread of the mature root system and are closely linked to the plant’s ability to handle stress. A good root structure is the basis for having enough root volume and surface area to function well.

To adapt to diverse environments, wheat varieties have accumulated numerous genetic variations. Genetic enhancement of crop roots has seldom been analyzed [14,15]. The identification of QTL provides an effective strategy to develop molecular markers and identify candidate genes [16]. A thorough understanding of the genetic foundation of drought root system architecture traits would aid in optimizing root systems under conditions of nutrient deficiency [2,3,5,6,7,8,12,13,14]. Several QTLs associated with root system architecture-related traits have been uncovered in wheat [17,18]. Root system architecture is a highly adaptable trait across various environments [22]; QTLs governing root system architecture under drought conditions are crucial for enhancing drought tolerance. Over 10 loci influencing the number of root tips under drought stress have been mapped on chromosomes 4A, 4B, 4D, 5B, 1D, 6D, and 7D [23]. The locus on chromosome 4D (360.3–396.8 Mb) is different from the loci identified in this study (*QDNRT.daas-4DS* 1.2–3.6 Mb). Thus, *QDNRT.daas-4DS* is a new loci. Over 10 loci for root surface area under drought stress were identified on chromosomes 1D, 2A, 2B, 2D, 3B, 4A, 5B, 5D, 7A, and 7D in common wheat [23]. Of these, the locus on 4A (565.6–598.2 Mb) is nearly adjacent to the *QDTRS.daas-4AL* (613.6–615.2 Mb).

In total, 12 loci for total root length under drought stress were identified on chromosomes 1A, 1D, 2A, 2D, 3A, 3B, 3D, 4B, 5B, 5D, 7A, and 7D [23,24,25]. In this study, we have identified a locus for total root length under drought stress, *QDTRL.daas-3AL* (650.4–659.4 Mb), which is near the loci on 3AL (632.8–646.3 Mb). Until now, eight loci for DDRW were identified on chromosomes 1B, 2A, 2D, 4A, 4B, 5A, 7A, and 7D [1,26,27,28,29]. We identified two loci for DDRW, *QDDRW.daas-1BL* (586.3–609.0 Mb) and *QDDRW.daas-5D* (378.9–393.5 Mb), which differ from the loci located on chromosome 1B (120.2–159.6 Mb) and 5D (489.6–540.3 Mb) mentioned above. Thus, both *QDDRW.daas-1BL* and *QDDRW.daas-5D* are novel.

The genes associated with plant height and vernalization may also have significant effects on root system architecture-related traits [30]. Over the past 80 years, several genetic loci associated with plant height and vernalization have been identified in common wheat, and a number of functional genes have been cloned on chromosome 1B (*Rht2*/*Rht10* at 19.18 Mb), 4D (*SVP3-4D/BM1-4D* at 469.46 Mb, *Vrn2-4D/ZCCT1-4D* at 509.43 Mb), and 5D (*TaDEP1-5D* at 329.11 Mb, *Rht23* at 524.96 Mb, and *Vrn1-5D* at 470.00 Mb) [31], and 6A (Rht24, 411.93–414.88 Mb) [32]. Based on physical positions, the loci *QDTRS.daas-4AL* (613.6–615.2 Mb) and *QDNRT.daas-4DS* (1.2–3.6 Mb), *QDTRL.daas-3AL* (650.4–659.4 Mb), *QDDRW.daas-1BL* (586.3–609.0 Mb), and *QDDRW.daas-5D* (378.9–393.5 Mb) are different from the reported plant height and vernalization genes.

Notably, compared with previous results and meta-analyses, *QDDRW.daas-1BL*, *QDNRT.daas-4DS*, and *QDDRW.daas-5D* were novel. We have presented the linkage mapping results for agronomic traits in the Zhoumai16/DK171 RIL population [15] and pinpointed several genomic regions linked to both drought root system architecture-related traits and agronomic traits. Specifically, *QDDRW.daas-1BL* (586.3–609.0 Mb) overlaps with a QTL cluster (556–654 Mb) influencing kernel number per spike (KNS), PH, and flag leaf width (FLW), and *QDDRW.daas-5D* (378.9–393.5 Mb) co-locates with a QTL cluster (277.0–491.2 Mb) related to KNS, FLW, TKW, and heading date [15]. These findings suggest that the loci associated with drought response and survival mechanisms may also serve as targets for enhancing yield potential and stability.

The limitation raised regarding the absence of control experiments in this study, which currently prevents definitive verification of whether the identified QTLs are specific to drought conditions. Although these QTLs demonstrated significant effects under drought stress, their potential functionality under non-stress conditions remains unexamined. Unfortunately, insufficient seed availability precluded the inclusion of control treatments in the present experiment. To address this gap, subsequent phenotyping of root-related traits will be conducted under optimal growing conditions. The acquired data will support two analytical approaches: direct QTL mapping under control conditions to compare with drought-induced QTLs, and mapping based on trait ratios between stress and control conditions. These analyses will help distinguish QTLs unique to drought response from those constitutive to plant growth—a distinction critical for breeding applications. While the current study provides directly relevant insights for drought tolerance breeding and genetic dissection under water-limited environments, further validation under controlled conditions will significantly enhance the biological interpretation and practical utility of these loci.

A total of eight candidate genes were identified, primarily implicated in the biological metabolism of plant hormones and calcium-dependent lipid-binding domain proteins. Among these, *TraesCS4D01G001900* (*QDNRT.daas-4DS*) encodes a CDPK-related kinase, playing a crucial role in diverse signaling pathways for root growth and development [33,34], such as root hair growth and cell length [35]. Additionally, *TraesCS1B01G373900*, associated with *QDDRW.daas-1BL*, encodes an ABC transporter family protein essential for primary root growth and shoot development. Root development is governed by various plant hormones [36]. *TraesCS4A01G326400* (*QDTRS.daas-4AL*) and *TraesCS4D01G002400* (*QDNRT.daas-4DS*) encode ethylene-responsive transcription factors [37]. Ethylene plays diverse roles in growth, development, signal transduction, and cell differentiation, including root growth [8,9], and influences drought root system architecture-related traits like root hair and cluster root formation [25]. *TraesCS1B01G356600* (*QDDRW.daas-1BL*), *TraesCS4D01G002400* (*QDNRT.daas-4DS*), and *TraesCS5D01G285900* (*QDDRW.daas-5D*) encode the auxin-responsive proteins, auxin transport protein, and auxin-induced protein 12 in root cultures. Auxin, a core regulator, integrates with other plant hormones to regulate root development. The biosynthesis, transport, and signaling pathways of auxin, particularly indole-3-acetic acid, are crucial for plant root development [20]. *TraesCS3A01G406200* (*QDTRL.daas-3AL*) encodes the gibberellin 20 oxidase 2 (GA20ox), a key enzyme in gibberellin synthesis [36,37], which regulate various stages of plant growth and development, promoting seed germination, plant growth, flowering induction, and other biological functions.

In this study, the candidate genes were preliminarily screened through bioinformatic annotation and expression profiling analyses. These candidate genes currently serve only as reference targets as their biological functions remain to be experimentally validated. To systematically characterize these candidate genes, the following research pipeline were applied: (1) construction of a secondary mapping population coupled with KASP marker development for high-resolution genetic mapping; (2) comprehensive identification of target genes through integrated transcriptomic and genomic variation analyses; (3) functional validation employing both gene editing (e.g., CRISPR/Cas9) and transgenic complementation approaches. It should be emphasized that the KASP markers utilized in this study were specifically designed as genetic linkage markers rather than functional markers.

Traditional wheat breeding primarily focuses on yield and disease-related traits, with root system architecture closely linked to yield traits. Although traditional breeding has improved root system characteristics, the selection process remains lengthy and less efficient due to the challenges in field measurement of drought root system architecture-related traits [19]. Moreover, seedling root development is crucial for early wheat growth. KASP markers have been widely adopted for detecting genetic variations in wheat, enabling high-throughput genotyping. By utilizing genotype data from wheat SNP arrays for QTL mapping and genome-wide association studies, linked SNPs can be converted into KASP markers, which can then be directly applied in marker-assisted selection breeding programs. This approach facilitates the efficient identification and selection of desirable traits in wheat breeding efforts [19]. KASP markers are extensively applied in the improvement of yield, disease resistance, and quality traits in wheat. In this study, we successfully developed based on tightly linked SNP markers. The accuracy rates of *Kasp_1BL_DTRS*, *Kasp_3AL_DTRS*, *Kasp_DTRS_4A*, *Kasp_DDRW_5D*, and *Kasp_DNRT_4D* in detecting superior and non-superior phenotypes for DTRS, DDRW, and DNRT were 66.0% and 75.5%; 90.6% and 92.5%; 60.4% and 73.6%; 58.5% and 79.2%; 52.8% and 66.0%, respectively. Thus, these KASP markers could be used as valuable tools in MAS breeding programs. Additionally, accessions carrying more favorable alleles and exhibiting superior DRSA traits along with desirable agronomic characteristics, such as Jinmai 61, Liangxing 99, Yumai 35, Yumai 47, Liangxing 66, Bainong 64, Lumai 8, Yanzhan 4110, Zhengmai 366, Jimai 22, and Aikang 58, are recommended as parental lines for the improvement of drought root system architecture traits.

## 4. Materials and Methods

### 4.1. Plant Materials and Phenotypic Traits

Zhoumai16 is a high-yield winter wheat variety in the Huang-Huai wheat region. It is suitable for high-fertilizer and high-water cultivation and has moderate drought tolerance. DK171 is a newly developed high-yield and stress-tolerant winter wheat variety, with higher drought tolerance. This study utilized a Zhoumai16/DK171 F_2:6_~ RIL population to conduct hydroponic experiments under greenhouse drought stress conditions, measuring DRSA-related traits with three replicates.

The standard Hoagland nutrient solution includes: Macronutrients (mg/L): Calcium Nitrate (Ca(NO_3_)_2_·4H_2_O) at 945 mg/L, Potassium Nitrate (KNO_3_) at 607 mg/L, Ammonium Dihydrogen Phosphate (NH_4_H_2_PO_4_) at 115 mg/L, Magnesium Sulfate (MgSO_4_·7H_2_O) at 493 mg/L. Micronutrients (mg/L): EDTA-Iron (Fe-EDTA) at 20 mg/L, Boric Acid (H_3_BO_3_) at 2.86 mg/L (or Borax), Manganese Sulfate (MnSO_4_·H_2_O) at 2.13 mg/L, Zinc Sulfate (ZnSO_4_·7H_2_O) at 0.22 mg/L, Copper Sulfate (CuSO_4_·5H_2_O) at 0.08 mg/L, and Ammonium Molybdate ((NH_4_)_6_Mo_7_O_24_·4H_2_O) at 0.02 mg/L. The solution pH should be adjusted to 5.5–6.5 using acid/base before use. Prepare 1× full-strength Hoagland solution in advance and adjust its pH to 5.8–6.0. Weigh 5.0 g, 7.5 g, 10.0 g, 12.5 g, 15.0 g, 17.5 g, and 20.0 g of PEG 6000 into seven clean beakers. Add about 60 mL of the pre-warmed (50–60 °C) 1× Hoagland solution to each beaker and stir gently on a magnetic stirrer until the PEG 6000 is completely dissolved. Transfer each solution to its corresponding 100 mL volumetric flask, bring the volume to the mark with additional 1× Hoagland solution, and mix thoroughly by inverting the flasks.

The root dry weights of Zhoumai16 plants subjected to 5%, 7.5%, 10%, 12.5%, 15%, 17.5%, and 20% PEG6000 were 0.0128 g, 0.0122 g, 0.0098 g, 0.0090 g, 0.0065 g, 0.0060 g, 0.0052 g, and 0.0026 g, respectively. A 12.5% concentration of PEG 6000 was added to the culture medium to simulate drought conditions. The methodology is as follows: 20 wheat seeds from each line were randomly selected and surface-sterilized with 10% H_2_O_2_ for 20 min, then placed in Petri dishes containing moist filter paper. When the coleoptile length reached approximately 2 cm, the seedlings were transferred to plastic trays (53 × 27 cm) with Hoagland nutrient solution supplemented with 12.5% PEG 6000 to induce osmotic stress.

These trays were then placed in a constant temperature culture room maintained at 25 °C, with 16 h light and 8 h darkness. After three weeks of growth in the greenhouse, seedling DRSA-related traits, including DDRW, DTRL, DTRA, and DNRT, were measured using the WinRHIZO software V1.0 (https://www.quantitative-plant.org/software/winrhizo, accessed on 24 June 2025) (Regent, Vancouver, BC, Canada). The specific method was as follows: thoroughly washed wheat roots were neatly arranged in a scanning tray, and scanned images were obtained using an Expression 11000XL scanner (Seiko Epson Corporation, Nagano Prefecture, Japan). The scanned images were analyzed using WinRHIZO software. Five plants were measured for each line, with three biological replicates.

This study used 108 varieties to validate the effects of KASP markers. The phenotypic values of DDRW, DTRL, DTRA, and DNRT for this validation population were also uniformly identified using the aforementioned method.

### 4.2. Genome Wide Linkage Mapping

The Zhoumai16/DK171 RIL population was genotyped using the wheat 90K SNP chip (CapitalBio Corporation, Beijing, China). The quality control followed: SNPs were set at missing data exceeding 20% or minor allele frequency (MAF) below 0.5. Filtered SNPs were classified into bin markers by IciMapping v4.2 [17]. Subsequently, the regression mapping algorithm in JoinMap v4.0 was used to calculate linkage distances for the obtained BIN markers and construct a higher-density linkage map. The successfully constructed linkage map has been previously reported by Wen et al. [18] and Li et al. [15]. Based on the constructed high-density SNP genetic map and the obtained root DRSA-related traits under drought, genome-wide linkage mapping was conducted by the inclusive composite interval mapping (ICIM-add) using IciMapping v4.1 [17]. The logarithm of odds (LOD) threshold for significant QTLs was determined to be 2.60 based on 1000 permutations. The physical positions were determined by IWGSC v1.0.

### 4.3. Identification of Candidate Genes for Drought-Related Traits

To identify the candidate genes associated with drought-related trait QTLs detected in the Zhoumai16/DK171 RIL population, high-confidence annotation genes within the LD block region surrounding each QTL peak SNP were extracted from the wheat IWGSC v1.0 [21]. Combining annotation, high-confidence genes with relevant annotated functions and differences in the coding regions between the two parents were screened as candidate genes. To investigate the expression patterns and identify candidate genes with notable transcription levels in seedlings or root tissues, we utilized the publicly accessible *Triticum aestivum* gene expression database [12] (http://wheat-expression.com/, accessed on 24 June 2025). After phenotyping, roots were sampled, and RNA was extracted from the root samples using the TRIzol method. cDNA was synthesized using the HiScript II cDNA Synthesis Kit. Primers for qRT-PCR were designed with Primer Premier 5.0. The reaction mixture consisted of 20 µL, including 2 µL of cDNA, 10 µL of ChamQ Universal SYBR qPCR Master Mix, and 0.4 µL of each primer. *TaActin*1 was used as an internal control to normalize the expression levels of different samples. The gene expression levels were analyzed using the 2^−ΔΔCT^ method. All assays were performed with two biological replicates and three technical replicates.

### 4.4. KASP Marker Development and Validation

For all loci, flanking SNPs were converted to KASP markers [19], designed using PolyMarker (http://www.polymarker.info/, accessed on 24 June 2025). The 384-well plates were analyzed on a PHERAstarplus SNP, and genotyping was conducted by KlusterCaller (LGC) (https://www.lgcstandards.com/, accessed on 24 June 2025) (London, UK). All developed KASP markers required genetic effect validation using 108 varieties mainly from the Yellow and Huai Wheat Region [20]. These 108 materials primarily include the main popularized varieties, key backbone parents, and representative lines from the Yellow and Huai River Valleys Wheat Zone.

## 5. Conclusions

In conclusion, this study highlights the critical role of drought-related root system architecture in enhancing wheat resilience to drought. By analyzing the Zhoumai16/DK171 RIL population, five QTLs associated with DRSA-related traits were identified, including novel loci *QDDRW.daas-1BL, QDNRT.daas-4DS*, and *QDDRW.daas-5D*. These QTLs explained 6.1% to 18.9% of the phenotypic variances, with favorable alleles contributed by both Zhoumai16 and DK171. Additionally, five KASP markers were developed and validated, providing valuable tools for MAS breeding. The identification of eight candidate genes further enriches the genetic resources for wheat drought tolerance. This research advances our understanding of the genetic basis of drought resistance and offers practical tools for breeding drought-tolerant wheat cultivars.

## Figures and Tables

**Figure 1 plants-14-03023-f001:**
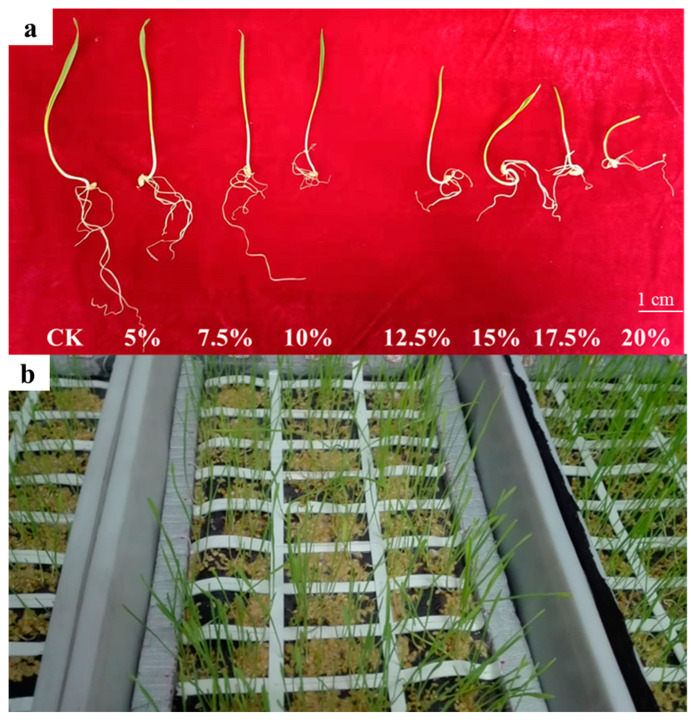
The DRSA-related traits in the Zhoumai16/DK171 RIL population. (**a**) Root growth of wheat seedlings under different concentrations. Root development status under different concentrations of PEG6000, with 12.5% ultimately selected as the drought condition for root phenotypic characterization (5-day). PEG-6000 was used to simulate drought stress. At concentrations of 5–10%, there was no significant reduction in germination potential or germination rate, while concentrations of 12.5–20% significantly inhibited germination. Based on the comparative analysis of relative germination rates and germination potentials at different concentrations, a concentration of 12.5% was selected for the assessment of seedling-related phenotypes. (**b**) Bulk phenotyping of wheat seedling root traits. Bulk evaluation of root phenotypes under 12.5% PEG6000 conditions (8-day growth status).

**Figure 2 plants-14-03023-f002:**
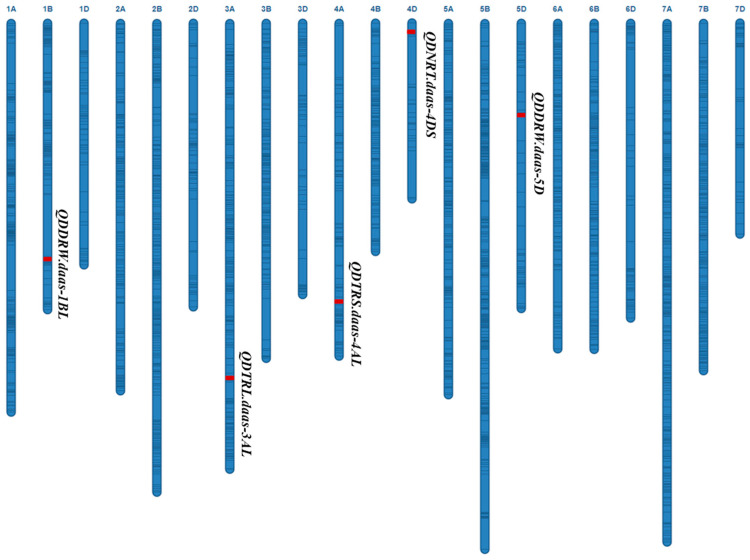
QTL for the DRSA-related traits in the Zhoumai16/DK171 RIL population.

**Figure 3 plants-14-03023-f003:**
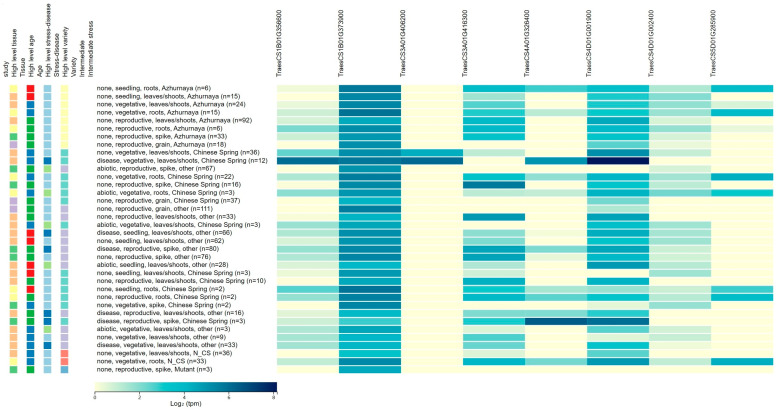
The expression patterns for the eight candidate genes associated with DRSA-related traits. The left side of the figure represents the expression patterns of candidate genes in different tissues and at various developmental stages. The data originates from wheat expression data. The specific transcriptome data can be downloaded from this website (http://wheat-expression.com/, accessed on 24 June 2025).

**Figure 4 plants-14-03023-f004:**
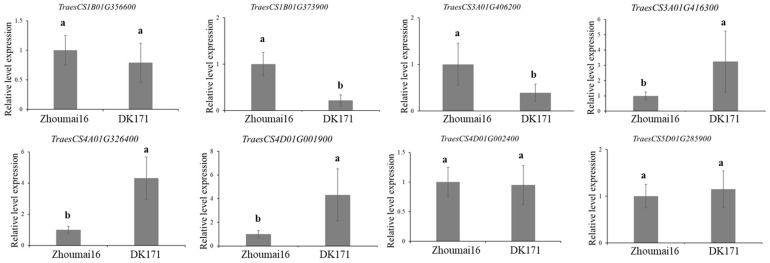
The qRT-PCR results for the candidate genes identified in this study. Transcriptional analysis was conducted in the RIL population of Zhoumai16 and DK171, where different letters indicate significant differences at the *p* < 0.05 level.

**Table 1 plants-14-03023-t001:** QTL for DRSA-related traits in Zhoumai16/DK171 RIL population.

QTL	Chromosome	Genetic Interval	Physical Position (Mb)	LOD	R^2^	Add
*QDDRW.daas-1BL*	1B	*wsnp_Ex_rep_c67299_65845319~* *Excalibur_rep_c107035_354*	586.3–609.0	11.1	18.9	−0.002
*QDTRL.daas-3AL*	3A	*Kukri_rep_c69970_717~Kukri_rep_c103783_1380*	650.4–659.4	3.0	6.1	−1.27
*QDTRS.daas-4AL*	4A	*IAAV7132~wsnp_JD_c38619_27992279*	613.6–615.2	3.5	9.6	0.28
*QDNRT.daas-4DS*	4D	*RAC875_rep_c76650_164~Kukri_c15720_884*	1.2–3.6	3.2	8.1	7.61
*QDDRW.daas-5D*	5D	*BobWhite_c5176_1164~RAC875_rep_c78046_324*	378.9–393.5	3.1	7.5	−0.001

**Table 2 plants-14-03023-t002:** The candidate genes for DRSA-related traits identified in the Zhoumai16/DK171 RIL population.

QTL	Candidate Gene	Chromosome	Start (bp)	Annotation
*QDDRW.daas-1BL*	*TraesCS1B01G356600*	1B	585539303	Auxin-responsive protein
*QDDRW.daas-1BL*	*TraesCS1B01G373900*	1B	604319205	ABC transporter A family protein
*QDTRL.daas-3AL*	*TraesCS3A01G406200*	3A	650429434	Gibberellin 20 oxidase 2
*QDTRL.daas-3AL*	*TraesCS3A01G416300*	3A	658983005	Auxin transport protein (BIG)
*QDTRS.daas-4AL*	*TraesCS4A01G326400*	4A	613543523	Ethylene-responsive transcription factor
*QDNRT.daas-4DS*	*TraesCS4D01G001900*	4D	1239483	Calcium-dependent lipid-binding domain protein
*QDNRT.daas-4DS*	*TraesCS4D01G002400*	4D	1288444	Ethylene-responsive transcription factor
*QDDRW.daas-5D*	*TraesCS5D01G285900*	5D	386209788	Auxin-induced in root cultures protein 12

**Table 3 plants-14-03023-t003:** Effects of *Kasp_4AL_DTRS*, *Kasp_4DS_DNRT*, and *Kasp_5D_DDRW* on DRSA-related traits in the natural population.

Marker Name	QTL	Genotype ^a^	Number of Lines	Phenotype	*p*-Value
*Kasp_1BL_DDRW*	*QDDRW.daas-1BL*	*AA*	74	6.80 (DTRS)	0.023 *
		GG	33	5.85 (DTRS)	
*Kasp_3AL_DTRL*	*QDTRL.daas-3AL*	*AA*	72	63.52 (DTRL)	0.010 *
		GG	35	56.60 (DTRL)	
*Kasp_4AL_DTRS*	*QDTRS.daas-4AL*	*GG*	60	6.99 (DTRS)	0.021 *
		AA	42	5.97 (DTRS)	
*Kasp_4DS_DNRT*	*QDNRT.daas-4DS*	*TT*	33	83.0 (DNRT)	0.009 *
		CC	71	97.7 (DNRT)	
*Kasp_5D_DDRW*	*QDDRW.daas-5D*	*CC*	11	0.0297 (DDRW)	0.039 *
		TT	96	0.0245 (DDRW)	

^a^ The italic is the favorable allele. * Significant at *p* < 0.05.

**Table 4 plants-14-03023-t004:** The primers of the KASP markers identified in this study.

Kasp Name	Primer	Sequence
*Kasp-1BL-DDRW*	FAM	GAAGGTGACCAAGTTCATGCTTTGAGGCGACCACCCTGA
	HEX	GAAGGTCGGAGTCAACGGATTTTGAGGCGACCACCCTGG
	COMMON	GCAGCCGTTATTCAACTTCTAA
*Kasp-3AL-DTRL*	FAM	GAAGGTGACCAAGTTCATGCTCCTTCTGGATTGATGGTTCTCA
	HEX	GAAGGTCGGAGTCAACGGATTCCTTCTGGATTGATGGTTCTCG
	COMMON	TCTGCCTCGAAGTCTTCATTT
*Kasp_4AL_DTRS*	FAM	GAAGGTGACCAAGTTCATGCTCATTGCCAAATGTTTGCTGTATT
	HEX	GAAGGTCGGAGTCAACGGATTCATTGCCAAATGTTTGCTGTATC
	COMMON	CATTATCAGATGATACCACGTCG
*Kasp_4DS_DNRT*	FAM	GAAGGTGACCAAGTTCATGCTTGAACTCGGCTGATACCAGA
	HEX	GAAGGTCGGAGTCAACGGATTGAACTCGGCTGATACCAGG
	COMMON	GGTGATGGCGAACCTAGAAAC
*Kasp_5D_DDRW*	FAM	GAAGGTGACCAAGTTCATGCTCATTGCCAAATGTTTGCTGTATT
	HEX	GAAGGTCGGAGTCAACGGATTCATTGCCAAATGTTTGCTGTATC
	COMMON	CATTATCAGATGATACCACGTCG

The underlined nucleotides are the FAM and HEX primers.

## Data Availability

All datasets generated for this study are included in the article; further inquiries can be directed to the first author.

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
