# Peer review of "Genome-Wide Linkage Mapping of Root System Architecture-Related Traits Under Drought Stress in Common Wheat (Triticum aestivum L.)"

_plants, 2025, doi:10.3390/plants14193023_

Round 1
Reviewer 1 Report
Comments and Suggestions for Authors
please refer to the attachment.

Author Response
The presented work meets the modern requirements of plant science. Genotyping and mapping are great tools for understanding the relationship between gene and trait, growth and development. Plants are complex organisms. And wheat is a complex object from the point of view of the genome. Therefore, such studies are important and necessary. Understanding the connection and implementation of genetic control and potential is relevant. Drought, which the authors modeled, is the predominant stress factor for crop production and limits growth and root development, which affects crop yields. The methods used in the work are adequate. The goal is clear. The authors have completed all the tasks set in the work, the results were obtained and discussed. Please pay attention. Please include my comments in the manuscript. This will improve its perception and make it more interesting and useful for the scientific community.
1 The title of the article confuses me. What does "common wheat" mean?
Response: Thank you very much for your suggestions. We fully understand your question regarding "common wheat." The genus Triticum includes multiple wheat species, such as common wheat (Triticum aestivum), durum wheat (Triticum durum), spelt wheat (Triticum spelta), and others. These species differ in terms of genomic structure, morphological characteristics, and uses. Common wheat (Triticum aestivum) is the most widely cultivated and important species within the Triticum genus. It possesses a complex hexaploid genome, exhibits strong adaptability, has high yields, and is widely used, making it the foundation of global grain production. Since common wheat is one of the world's major food crops, studying its genetics, physiology, and adaptability is of great significance for improving grain production and ensuring food security. Therefore, we emphasized "common wheat" in the title to highlight the importance and practical value of this research.
2 lines 62-66. These data are scanty. I ask you to provide more detailed information. Because this is what will help justify the purpose of your work. You should not focus only on the roots in this part of the work. You can consider what is generally known in the trait-gene relationship.
Response: Accepted. We sincerely apologize for the lack of clarity in our explanation. Here, we intended to convey that current research on wheat root systems under drought conditions is limited, and only a few loci have been reported so far. Since the focus of this paper is on the development of wheat roots under drought stress, we have only listed some of the currently reported root-related loci. Additionally, to reduce the length of the introduction section, we have included some research progress in the discussion section. We kindly ask you to review this part. Once again, we greatly appreciate your valuable suggestions for improving the manuscript.
3 Fig. 1. Please provide the age of the plants. Provide information on the condition of the roots as clearly as possible. You are focusing on the roots of the plants. Provide data on their length, wet and dry weight.
Response: Accepted. We are particularly grateful for your suggestions. We have added the time of root phenotype identification for the strains in Figure 1. During the exploration of phenotypic identification conditions, we did not measure the root length and fresh weight of these strains. However, we measured the dry weight (the dry weights corresponding to 5%, 7.5%, 10%, 12.5%, 15%, 17.5%, and 20% PEG6000 concentrations were 0.0128g, 0.0122g, 0.0098g, 0.0090g, 0.0065g, 0.0060g, 0.0052g, and 0.0026g, respectively). We have provided the dry weight data in the Materials and Methods section. Please kindly review it. We sincerely appreciate your assistance and guidance.
4 Consider the development. This will greatly improve the perception of the material. In addition, this will improve the Discussion section. In the discussion section, it is necessary to provide a link between the development of roots and candidate genes, etc. Here are exactly 21 day-old plants (line 280). This is already the pre-tillering stage. Specify the type of plants. Table S1 it is confusing. It is difficult to understand.
Response: We sincerely appreciate your assistance. Regarding the rationale for selecting 21-day-old seedlings, our considerations are as follows: (1) Critical Developmental Stage: The 21-day-old seedling stage (three-leaf to pre-tillering phase) represents a golden window for wheat root system architecture establishment. By this stage, the primary root system has fully expanded, and seminal roots begin to differentiate. Key traits such as root-to-shoot ratio and lateral root density already reflect early drought adaptation. (2) Phenotypic Detectability and Experimental Feasibility: At this stage, the root biomass is moderate, facilitating high-throughput sampling and precise measurements (e.g., WinRHIZO scanning analysis). (3) Optimal Timing for Genetic Analysis: Studies have reported that drought-responsive genes (e.g., DREB and NAC transcription factor families) in 21-day-old wheat roots reach peak expression levels under stress conditions. Conducting QTL mapping or GWAS analysis at this stage significantly enhances the association signals between root traits and candidate genes (Ren et al., 2012; Adeleke et al., 2020; Saddiq et al., 2021; Li et al., 2023).
Regarding the selection of candidate genes, we primarily considered the following aspects: (1) Selection of high-confidence candidate genes within the identified genetic intervals; (2) Based on existing reports, wheat root development may be associated with the following candidate genes; (3) Final determination of candidate genes by analyzing their expression patterns using public expression databases; (4) Screening the candidate gene list by combining RT-PCR results. Given the enormous size of the wheat genome (16 Gb), conventional QTL mapping cannot precisely identify the target genes. Therefore, we will proceed with the following steps: (1) Constructing derived near-isogenic line populations; (2) Developing KASP markers for fine mapping; (3) Defining the target intervals and preliminarily identifying candidate genes using RNA-seq and other methods; (4) Validating candidate genes through transgenic or gene-editing approaches. We have supplemented the relevant content in the discussion section for your reference.
Table S1 contains the phenotypic values for each line in the Doumai/Shi4185 RIL population. We acknowledge that our initial description was unclear. We have revised it and renamed the lines as Line1-262. We sincerely appreciate your assistance.
Ren Y, He X, Liu D, Li J, Zhao X, Li B, Tong Y, Zhang A, Li Z. Major quantitative trait loci for seminal root morphology of wheat seedlings. Molecular Breeding. 2012 Jun;30(1):139-48.
Adeleke, E., Millas, R., McNeal, W., Faris, J. and Taheri, A., 2020. Variation analysis of root system development in wheat seedlings using root phenotyping system. Agronomy, 10(2), p.206.
Li, L., Li, T., Liu, Y., Li, L., Huang, X. and Xie, J., 2023. Effects of antibiotics stress on root development, seedling growth, antioxidant status and abscisic acid level in wheat (Triticum aestivum L.). Ecotoxicology and environmental safety, 252, p.114621.
Saddiq, M.S., Iqbal, S., Hafeez, M.B., Ibrahim, A.M., Raza, A., Fatima, E.M., Baloch, H., Jahanzaib, Woodrow, P. and Ciarmiello, L.F., 2021. Effect of salinity stress on physiological changes in winter and spring wheat. Agronomy, 11(6), p.1193.
5 Specify what you measured? Line 285.
Response: Accepted. We are particularly grateful for your suggestions. We apologize for the lack of clarity in our writing. The specific method we used involved selecting 20 plump seeds and cultivating seedlings under 12.5% PEG6000 conditions. From these, we chose 5 plants with uniform growth for the evaluation of root system development-related phenotypes. We conducted three biological replicates and took the average for subsequent phenotypic studies. We have further revised the description to make it more detailed. We sincerely appreciate your assistance and guidance.
6 Specify at what level of stress you studied the plants. Or was it a general sample. Here the stress is variable.
Response: We are particularly grateful for your valuable suggestions. We apologize for the lack of clarity in our writing, which may have caused confusion. Initially, we used different concentrations of PEG6000 to determine the optimal concentration for simulating drought conditions (Figure 1). Based on the results from Figure 1 and considering the development of wheat roots, we ultimately selected a 12.5% concentration of PEG6000 to simulate drought conditions. The 12.5% concentration of PEG6000 is also a commonly used concentration for drought simulation (Siala et al., 2016; Li et al., 2020; Havii et al., 2025). Once again, thank you for your guidance.
Siala, M.A., Buxb, H., Mangriob, S.M., Memonb, H.M.U., Dahotc, M.U., Mujtabaa, S.M. and Sahitod, J.G.M., 2016, November. Early growth behaviour of wheat genotypes as affected by polyethylene glycol (PEG-6000). In Series B: Biological Sciences (p. 172).
Li, D., Batchelor, W.D., Zhang, D., Miao, H., Li, H., Song, S. and Li, R., 2020. Analysis of melatonin regulation of germination and antioxidant metabolism in different wheat cultivars under polyethylene glycol stress. PLoS One, 15(8), p.e0237536.
Havii, V., Palyvoda, Y., Kuchmenko, O., Stamirowska-Krzaczek, E., Tomaszewska, M. and Kocira, A., 2025. Biochemical mechanisms of drought resistance in soft wheat under modeling of water deficiency and effects of seed treatment with metabolically active substances. Agricultural Engineering, 29.
7 Good observation - lines 22-224.
Response: Accepted. We are particularly grateful for your suggestions and positive feedback. We investigated the development of wheat roots under drought conditions and compared the genetic loci related to roots with previously discovered loci associated with yield and agronomic traits. We found that some loci are correlated with both root development and yield, which can provide highly practical references for breeding high-yield and stable-yield wheat varieties. Once again, thank you for your valuable suggestions.
8 Lines 261-265 - good forecast.
Response: Accepted. We are particularly grateful for your suggestions and positive feedback. In addition to providing new markers, we aim to offer new and usable germplasm information to serve as a reference for wheat crossbreeding, thereby accelerating the process of breeding high-yield and stable-yield wheat varieties. Once again, thank you for your valuable suggestions on enhancing the quality of the article.
9 The Discussion part is enough.
Response: Accepted. We are particularly grateful for your positive feedback. We have further optimized the discussion section, striving to minimize grammatical and textual errors and enhance the logical flow of the article.
Reviewer 2 Report
Comments and Suggestions for Authors
The study by Jin et al. investigates root architecture in bread wheat under drought stress. Identifying genetic determinants regulating drought tolerance is highly relevant for plant breeding and agriculture, particularly in the context of climate change, freshwater scarcity, and the need to utilize marginal lands. The authors employed F2:6 recombinant inbred line (RIL) population derived from a cross between the Zhoumai 16 and DK171 varieties. Phenotypic analysis was conducted on 262 RILs, with 20 seeds per line, at the seedling stage under simulated drought conditions using 12.5% PEG 6000. Root length, surface area, root number, and dry mass were assessed as key phenotypic traits. Genome-wide association study (GWAS) analysis identified five quantitative trait loci (QTLs) associated with these traits, and KASP markers were developed for the corresponding polymorphisms (two for dry mass, and one each for root length, number, and surface area). Based on the identified QTLs, eight candidate genes were proposed, and differential expression between the parental lines was demonstrated.
However, several methodological concerns should be addressed:
1. The experiment should include a control treatment with Hoagland nutrient solution (without PEG 6000) for phenotypic evaluation of RILs.
2. Similarly, gene expression analysis in parental lines should incorporate a control (Hoagland solution without PEG 6000).
3. The study would benefit from assessing candidate gene expression in RILs, particularly by comparing contrasting genotypes in response to PEG 6000.
4. The experimental design appears unbalanced: root traits were measured in RILs, whereas gene expression was analyzed only in parental lines. Including parental forms in root phenotyping would strengthen the study.
5. The methodology for drought treatment in gene expression analysis is unclear. Key details—such as growth stage, treatment conditions (PEG 6000 vs. Hoagland solution), and biological/technical replication—should be specified.
6. No polymorphisms were reported in candidate gene sequences between parental lines. Identifying such differences would enhance the reliability of KASP markers, ensuring they reflect functional rather than merely linked polymorphisms.
7. The growth habit (winter/spring) of Zhoumai 16 and the resulting RILs should be clarified, particularly if DK171 is a winter wheat variety.
8. Information on the drought tolerance of Zhoumai 16 would provide useful context.
9. Linkage decay analysis based on RIL genotyping data should be presented.
Technical corrections:
- Figure 1: Provide a clear, descriptive title rather than a results summary. Subpanels should be labeled (a, b, etc.) with corresponding subtitles.
- Units: Use superscript formatting (e.g., cm² instead of cm2) (Lines 80, 154, Table S1 header).
- Duplication: Remove repeated "China" (Line 269).
- Italics: Gene and species names should be italicized (e.g., TaActin1 (Line 320), Triticum aestivum (Line 315)).
Author Response
The study by Jin et al. investigates root architecture in bread wheat under drought stress. Identifying genetic determinants regulating drought tolerance is highly relevant for plant breeding and agriculture, particularly in the context of climate change, freshwater scarcity, and the need to utilize marginal lands. The authors employed F2:6 recombinant inbred line (RIL) population derived from a cross between the Zhoumai 16 and DK171 varieties. Phenotypic analysis was conducted on 262 RILs, with 20 seeds per line, at the seedling stage under simulated drought conditions using 12.5% PEG 6000. Root length, surface area, root number, and dry mass were assessed as key phenotypic traits. Genome-wide association study (GWAS) analysis identified five quantitative trait loci (QTLs) associated with these traits, and KASP markers were developed for the corresponding polymorphisms (two for dry mass, and one each for root length, number, and surface area). Based on the identified QTLs, eight candidate genes were proposed, and differential expression between the parental lines was demonstrated.
However, several methodological concerns should be addressed:
- The experiment should include a control treatment with Hoagland nutrient solution (without PEG 6000) for phenotypic evaluation of RILs.
Response: We are particularly grateful for your suggestion. There are two methods to analyze the genetic mechanisms of root growth in wheat under drought conditions. Generally, there are two approaches: measuring the ratio of traits under drought conditions to those under normal conditions, and directly measuring traits under drought conditions. Using the ratio method as the phenotype, the core objective is to identify QTLs that control the plant's ability to respond to drought stress (drought tolerance), i.e., the genotype's ability to maintain relative growth under stress. On the other hand, directly using phenotypes under drought conditions aims to identify QTLs that directly determine the absolute growth performance of roots in drought-stressed environments. The key advantage of the ratio method lies in standardizing the inherent growth differences among genotypes under normal conditions. It focuses on the relative rate of change induced by stress, reduces background variation interference, and more purely reflects drought tolerance responses. In contrast, the results of the direct method incorporate both the genotype's drought tolerance and its potential growth performance under ideal conditions. For example, longer roots under drought could stem from strong drought tolerance or simply from the genotype's inherent vigorous growth potential. The ratio method is more suitable for in-depth analysis of drought adaptation mechanisms and for breeding varieties that perform stably in fluctuating or intermittent drought environments (where the target trait is drought tolerance itself). The direct method is more applicable for direct phenotypic selection breeding in target drought environments (such as regions with persistent drought), where the goal is performance under specific drought conditions. Since this study leans more toward direct phenotypic selection breeding for target drought environments, we chose to conduct the research using phenotype data directly from drought conditions. We will subsequently measure root-related phenotypes under normal conditions to study the genotype's ability to maintain relative growth under stress. Many similar studies have also used a single PEG condition for genetic analysis (Zaidi et al., 2016; Ayalew et al., 2018; Guo et al., 2020). Also, we will arrange an additional treatment without Hoagland nutrient solution for root phenotype studies and will use a ratio-based approach for further genetic analysis.
Reference
Guo, J., Li, C., Zhang, X., Li, Y., Zhang, D., Shi, Y., Song, Y., Li, Y., Yang, D. and Wang, T., 2020. Transcriptome and GWAS analyses reveal candidate gene for seminal root length of maize seedlings under drought stress. Plant Science, 292, p.110380.
Ayalew, H., Liu, H., Börner, A., Kobiljski, B., Liu, C. and Yan, G., 2018. Genome-wide association mapping of major root length QTLs under PEG induced water stress in wheat. Frontiers in Plant Science, 9, p.1759.
Zaidi, P.H., Seetharam, K., Krishna, G., Krishnamurthy, L., Gajanan, S., Babu, R., Zerka, M., Vinayan, M.T. and Vivek, B.S., 2016. Genomic regions associated with root traits under drought stress in tropical maize (Zea mays L.). PloS one, 11(10), p.e0164340.
- Similarly, gene expression analysis in parental lines should incorporate a control (Hoagland solution without PEG 6000).
Response: Accepted. We also strongly agree with your suggestion. This study aims to identify QTLs that directly determine the absolute growth performance of roots in drought-stressed environments. Therefore, we focused more on comparisons between different materials (extreme root phenotypes) and did not include a control treatment. In our next study, we will set up a control treatment to further investigate the genetic mechanisms of drought tolerance.
- The study would benefit from assessing candidate gene expression in RILs, particularly by comparing contrasting genotypes in response to PEG 6000.
Response: Accepted. We fully agree with your suggestion. However, given the large number of lines in the RIL population, we considered two approaches: one was to compare expression differences using the parental lines, and the other was to form resistant and susceptible pools using extreme materials from the RIL lines. Since the background of the extreme materials in the resistant and susceptible pools was relatively complex and the results were less clear, we ultimately opted to use the parental lines for comparison. In the further study, we will first construct a near-isogenic line (NIL) population to further refine the genetic interval (to approximately 1.0 Mb). Additionally, we will conduct transcript expression analysis using multiple extreme resistant and susceptible pools to further identify candidate genes.
- The experimental design appears unbalanced: root traits were measured in RILs, whereas gene expression was analyzed only in parental lines. Including parental forms in root phenotyping would strengthen the study.
Response: Accepted. Thank you for your suggestion. We have conducted root-related phenotyping of the parental lines and have supplemented these data in the phenotypic results section. We sincerely appreciate your advice, which is highly pertinent to elucidating the genetic mechanisms of drought tolerance in wheat roots. Your insightful comments have significantly contributed to improving the quality of this manuscript.
- The methodology for drought treatment in gene expression analysis is unclear. Key details—such as growth stage, treatment conditions (PEG 6000 vs. Hoagland solution), and biological/technical replication-should be specified.
Response: Accepted. Thank you for your reminder. We have added detailed information regarding growth stage, treatment conditions, and biological/technical replication, and have supplemented this in the Materials and Methods section.
- No polymorphisms were reported in candidate gene sequences between parental lines. Identifying such differences would enhance the reliability of KASP markers, ensuring they reflect functional rather than merely linked polymorphisms.
Response: We sincerely appreciate your valuable suggestions. In this study, the candidate genes were preliminarily screened through bioinformatic annotation and expression profiling analyses. These candidate genes currently serve only as reference targets, as their biological functions remain to be experimentally validated. To systematically characterize these candidate genes, we propose the following research pipeline: Construction of a secondary mapping population coupled with KASP marker development for high-resolution genetic mapping; Comprehensive identification of target genes through integrated transcriptomic and genomic variation analyses; Functional validation employing both gene editing (e.g., CRISPR/Cas9) and transgenic complementation approaches. It should be emphasized that the KASP markers utilized in this study were specifically designed as genetic linkage markers rather than functional markers. We anticipate that these research findings may provide molecular tools and theoretical foundations for improving drought tolerance in wheat breeding programs.
- The growth habit (winter/spring) of Zhoumai 16 and the resulting RILs should be clarified, particularly if DK171 is a winter wheat variety.
Response: Zhoumai 16 is a winter wheat variety widely cultivated in Henan and Shandong provinces. DK171 is another winter wheat variety known for its good yield performance and superior drought tolerance. To exploit the high-yield loci from Zhoumai 16 and drought tolerance loci from DK171, we developed the Zhoumai 16/DK171 population. We have included the relevant information in the Materials and Methods section as well as the Results and Discussion section.
- Information on the drought tolerance of Zhoumai 16 would provide useful context.
Response: Accepted. Zhoumai 16 is a high-yielding variety, but its drought tolerance is slightly weaker than that of DK171. We have supplemented the drought tolerance-related information in the Materials and Methods section.
- Linkage decay analysis based on RIL genotyping data should be presented.
Response: Many thanks for your valuable suggestions. In the manuscript, we did not conduct LD (Linkage Disequilibrium) decay analysis on the RIL (Recombinant Inbred Line) population for the following reasons: (1) The RIL population was subjected to linkage analysis for genetic dissection, where the identified loci represent confidence intervals. The genetic intervals resolved by QTL mapping are likely more precise than those inferred from LD decay analysis. (2) The RIL population has undergone multiple generations of recombination, resulting in significantly more effective recombination events compared to natural populations. In advanced-generation RILs, LD levels are greatly reduced, with decay distances being extremely short (often limited to adjacent markers). Consequently, the results cannot reflect the true LD structure of natural or breeding populations in wheat, rendering them biologically uninformative. (3) RILs are derived from biparental crosses, exhibiting a highly homogeneous genetic background. They lack key factors that shape LD in natural populations, such as population substructure, historical admixture, or selection pressure. Instead, their LD patterns are primarily determined by physical distance and artificial recombination.
Technical corrections:
- Figure 1: Provide a clear, descriptive title rather than a results summary. Subpanels should be labeled (a, b, etc.) with corresponding subtitles.
Response: Accepted. We have corrected it in the revised version. Many thanks for your valuable suggestions.
- Units: Use superscript formatting (e.g., cm² instead of cm2) (Lines 80, 154, Table S1 header).
Response: Accepted. Many thanks for your kindly reminder.
- Duplication: Remove repeated "China" (Line 269).
Response: Accepted. Many thanks for your kindly reminder.
- Italics: Gene and species names should be italicized (e.g., TaActin1 (Line 320), Triticum aestivum (Line 315)).
Response: Accepted. We have revised it across all the whole manuscript.
Reviewer 3 Report
Comments and Suggestions for Authors
I read with interest this manuscript, it has good value but needs to be polished to not only show that what is stated in the manuscript is valid. Probably can be resolved by describing better some portions and add more into the conclusion. Suggestions are in the attached document.

The language can be polished for better clarity.
Author Response
Reviewer 2 Overall:
- The abstract language must be improved. I would re-write the full abstract.
Response: Accepted. We sincerely appreciate your suggestions. We have rewritten the abstract section, correcting the content, word choice, and grammatical errors.
- Improve the content of each figure legend. The description is very incomplete.
Response: Accepted. Thank you for your suggestion. We have supplemented the legends for all the figures, and they are now included in the revised manuscript. Please review them at your convenience.
Specific:
- Figure 1. Must add after how many days the pictures were taken
Response: Accepted. We have added the details for the pictures in the revised version, including the Fig. 1 and the M&M section.
- On table 2 please add the indication of the supplementary file with all data obtained,
including probabilities
Response: Many thanks for your valuable suggestions. Regarding the candidate genes listed in Table 2: these were identified based on annotation information and transcript analysis by public database (http://wheat-expression.com/). It is important to note that we cannot assign definite probabilities to individual candidate genes at this stage. Pinpointing specific causal genes directly within the highly complex wheat genome is nearly impossible based solely on initial annotation and transcript data. Candidate genes represent our best initial targets derived from these analyses. Our next steps involve integrating fine-mapping, detailed transcriptome analysis, and functional validation studies to ultimately identify and verify the true target gene(s) responsible for the observed phenotype. We have added the relevant section of the manuscript to explicitly clarify this context.
- Figure 4 add the probability and significance in each graph bar
Response: Accepted. We have added the probability and significance in each graph bar in the revised version.
- For the favorable alleles found add as supplementary file the image for each KASP primers with indication of the 3 different groups (homozygous for each allele and heterozygous), this will be informative for people that want to use the primers described in the manuscript.
Response: Accepted. Thank you very much for your valuable suggestion. We have resubmitted the images for these markers. We provide a supplementary file featuring images for each KASP primer, clearly indicating the three distinct groups: homozygous for each allele and heterozygous.
- In QTL validation it is not described in which materials the validation was done, and why these lines were selected. Please provide the materials info where the markers were tested as supplementary data.
Response: Accepted. Our initial study utilized a complete panel of 166 accessions (collected in 2011 and cultivated across multiple environments during 2012-2014) (Liu et al., 2017). This collection represented the predominant cultivars from the Huang-Huai wheat region, supplemented with selected international germplasms. Recent observations revealed progressive genetic deterioration within this population, resulting in increased heterogeneity. Consequently, we refined our selection to 108 genetically stable accessions for subsequent investigations.
- In the statement: ‘In this study, we successfully developed Kasp_1BL_DTRS (QDDRW.daas-1BL), Kasp_3AL_DTRS (QDTRL.daas3AL), Kasp_DTRS_4A for DTRS, Kasp_DDRW_5D for DDRW, and Kasp_DNRT_4D for DNRT based on tightly linked SNP markers, demonstrating their effectiveness as valuable tools in MAS breeding programs.’- please give a real example, otherwise must re-adjust the sentence.
Response: Accepted. We have revised the statement as the original wording was indeed inappropriate, and we have made the necessary modifications in the revised veision.
- Give the reference for plant materials
Response: Accepted. We have added the reference for the plant materials in the revised version. (Liu et al., 2017).
Liu, J., He, Z., Rasheed, A., Wen, W., Yan, J., Zhang, P., Wan, Y., Zhang, Y., Xie, C., & Xia, X. (2017, November 23). Genome-wide association mapping of black point reaction in common wheat (Triticum aestivum L.). BMC Plant Biology, 17(1), 220. https://doi.org/10.1186/s12870-017-1211-9017)
- Describe better the technique of KASP-marker selection and validation. This portion should be separated from the GW linkage mapping portion. Indicate all info (city and website) regarding LGC. Add as supplementary data the graphs of the validated KASP markers. In material and methods, this portion of the KASP genotyping should come after the portion of gene identification.
Response: We sincerely appreciate your valuable suggestions. In the revised manuscript, we have made the following modifications: (1) Provided more detailed methods for KASP marker selection and validation; (2) Separated the KASP marker development/validation section from the linkage analysis section; (3) Included all relevant information about LGC (company location and website);
(4) Submitted the validated KASP marker profiles as supplementary data (Table S3); (5) Reorganized the structure by placing the KASP genotyping content after the gene identification section; (6) Made appropriate additions to other materials and methods sections as well.
- Shouldn’t the conclusion come after discussion? Authors mut follow the directions of the journal formatting.
Response: Accepted and we have corrected it in the new version.
Response: Accepted. We have revised it in the new version.
- In the discussion and conclusion add the marker accuracy for the traits under drought stress. Be very specific on the traits and not overall without presenting data and real examples.
Response: Accepted. Thank you very much for your suggestion. We believe your suggestions are highly valuable. During the drafting process, we considered how to demonstrate the usability and reliability of the markers to better provide reference data for readers. In the initial draft, we employed T-tests for different alleles to prove the linkage between markers and target traits. For individual markers, using the accuracy rate of phenotypic correlation provides a more intuitive and significant measure of the detection accuracy. We have supplemented the content related to accuracy rates in both the results and discussion sections. Please review the specific details organized below. Once again, thank you for your valuable suggestions to enhance the quality of the manuscript.
To assess the accuracy of markers in detecting root phenotypes in natural populations, we selected the median value as the threshold for each trait. Phenotype values below the median were categorized as non-superior phenotypes, while those above were classified as superior phenotypes. We calculated the consistency rate between genotypes and phenotypes to provide a reference for the detection and evaluation of wheat root systems under drought conditions. The accuracy rates of Kasp_1BL_DTRS, Kasp_3AL_DTRS, Kasp_DTRS_4A, Kasp_DDRW_5D, and Kasp_DNRT_4D in detecting superior and non-superior phenotypes for DTRS, DDRW, and DNRT were 66.0% and 75.5%; 90.6% and 92.5%; 60.4% and 73.6%; 58.5% and 79.2%; 52.8% and 66.0%, respectively.
- Table S1 - indicate the nomenclature of abbreviations, e.g. DS l. Please replace ‘totally’ by a better synonymous word
Response: Accepted. In the revised version, we have made modifications to the abbreviations and all the whole text. DS is the designation for the Zhoumai 16/DK171 population used in this study. To avoid any confusion for the readers, we have revised this to Line1-Linenn here.
- In discussion write the full form of RSA - it is difficult for colleagues that are not in the area to remember what is RSA and have to go again to the beginning of the manuscript to know what is it
Response: Accepted. We have revised it in the new version and use the full form of RSA in Discussion and Conclusion section.
We sincerely appreciate your valuable suggestions for the manuscript. Your comments have significantly improved the quality of the paper. In addition to the specific recommendations you provided, we have also revised the grammar, word , and logical flow throughout the entire text.
Round 2
Reviewer 1 Report
Comments and Suggestions for Authors
The authors have done a great job. Thank you. My comments and remarks have been taken into account as of now.
Author Response
Response to Reviewer 1
Reviewer 1: The authors have done a great job. Thank you. My comments and remarks have been taken into account as of now.
Response: We deeply appreciate your positive feedback and are grateful for your recognition of our work. Your suggestions have significantly enhanced the language and content of the manuscript, and we are deeply grateful for your valuable insights.
Reviewer 2 Report
Comments and Suggestions for Authors
Dear Authors,
Thank you for your detailed responses and revisions to the manuscript. In many respects, the manuscript has indeed improved compared to the original version. However, several critical concerns remain unresolved:
1 and 2. You provided references to studies purportedly employing a "direct method" without controls. However, upon closer examination, these studies did include control treatments:
Guo et al. (2020): "under well-watered (WW) and water-stressed (WS) conditions."
Ayalew et al. (2018): "Hoagland’s solution alone for the control (non-stressed) set, respectively."
Zaidi et al. (2016): "thereafter, irrigation was stopped in cylinders labeled for drought stress treatment, whereas application of a measured amount of water was continued in cylinders labeled for well-watered treatment."
From a methodological standpoint, a control experiment is essential to filter out SNPs not specifically associated with drought stress. Currently, the study lacks evidence that the identified QTLs are exclusive to drought conditions. While they manifest under stress, their absence under optimal conditions remains unverified. Thus, experiments including controls for both parental lines and RILs are necessary.
- Accepted. It would be appropriate to integrate this part of your response into the Discussion section, as it highlights the limitations of the current findings and future research directions.
- While phenotypic measurements for parental lines were provided, statistical significance of differences in the three traits (DTRL, DTRS, DNRT) must be explicitly demonstrated.
- PEG 6000 solution preparation protocol (Line 537): The phrase "The dry weights for plants (Zhoumai16) corresponding to…" appears erroneous and requires correction.
- Accepted. As with point 3, this clarification should be added to the Discussion to contextualize the results’ limitations.
7a. Nomenclature consistency: The cultivar name alternates between "Zhoumai 16" (with space) and "Zhoumai16" (without space). Uniform formatting must be applied throughout the text.
7b. Parental line description: While additional details were added, the text still omits clarification that Zhoumai 16 is a winter wheat cultivar.
- Accepted.
- Accepted.
- Figure 1a: A scale bar must be added to the seedling photograph for reference.
Author Response
Response to Reviewer 2
Reviewer 2: Thank you for your detailed responses and revisions to the manuscript. In many respects, the manuscript has indeed improved compared to the original version. However, several critical concerns remain unresolved:
1 and 2. You provided references to studies purportedly employing a "direct method" without controls. However, upon closer examination, these studies did include control treatments: Guo et al. (2020): "under well-watered (WW) and water-stressed (WS) conditions."Ayalew et al. (2018): "Hoagland’s solution alone for the control (non-stressed) set, respectively."Zaidi et al. (2016): "thereafter, irrigation was stopped in cylinders labeled for drought stress treatment, whereas application of a measured amount of water was continued in cylinders labeled for well-watered treatment."
From a methodological standpoint, a control experiment is essential to filter out SNPs not specifically associated with drought stress. Currently, the study lacks evidence that the identified QTLs are exclusive to drought conditions. While they manifest under stress, their absence under optimal conditions remains unverified. Thus, experiments including controls for both parental lines and RILs are necessary.
Response: We extend our sincere gratitude to the reviewers for their meticulous review and invaluable suggestions. We fully concur with your perspective on the importance of control experiments and acknowledge the limitation of not explicitly setting up control experiments in our study.
- Regarding the cited reference (Guo et al., 2020; Ayalew et al., 2018; Zaidi et al., 2016), we understand that these studies indeed included control treatments. When citing these works, our intention was to illustrate the feasibility of direct methods, but we failed to sufficiently emphasize the critical role of control experiments in screening SNPs specifically related to drought stress. For this oversight, we deeply apologize.
- We fully agree with your observation that the absence of control experiments prevents us from fully verifying whether the identified QTL are exclusive to drought conditions. Although these QTL showed significant performance under stress conditions, their functionality under normal conditions remains unverified.
- Due to insufficient seed availability for experiments under normal conditions, we plan to propagate seeds in Hebei and Henan provinces of China this year. Upon harvest in 2026, we will conduct subsequent root-related trait measurements under control conditions. Post-data acquisition, we will perform linkage analysis in two ways: firstly, by directly using root traits under standard nutrient solution conditions for QTL mapping, comparing the two sets of QTL results to identify common and differential loci; secondly, by using the ratios of the two conditions as phenotypic values for QTL mapping.
We strongly endorse your suggestion that your recommendations can more effectively pinpoint which QTLs are unique to drought conditions (more responsive to drought stress) and which are common to both conditions (more likely related to plant growth), which is highly valuable for breeding. While our current study lacks data from standard nutrient solutions, we believe it still holds significance in wheat drought tolerance breeding. The results of this study provide direct feedback in drought environments, offering reference value for drought breeding and genetic mechanism analysis. Considering these points, we hope you will grant us the opportunity for our loci and markers to serve as references for wheat breeding in drought-prone areas. Once again, we thank you for your guidance and assistance, and we hope our subsequent supplementary work will meet your evaluation standards. Your suggestions have significantly enhanced the language and content of the manuscript, and we are deeply grateful for your valuable insights.
Comment 3: Accepted. It would be appropriate to integrate this part of your response into the Discussion section, as it highlights the limitations of the current findings and future research directions.
Response: We have incorporated the relevant content into the Discussion section to highlight the limitations and future research directions.
Comment 4: While phenotypic measurements for parental lines were provided, statistical significance of differences in the three traits (DTRL, DTRS, DNRT) must be explicitly demonstrated.
Response: We have added the statistical analysis (t-test results) to compare the parental lines of DTRL, DTRS and DNRT.
Comment 5: PEG 6000 solution preparation protocol (Line 537): The phrase "The dry weights for plants (Zhoumai16) corresponding to…" appears erroneous and requires correction.
Response: We apologize for the unclear expression. The sentence has been revised to:
“The root dry weights of Zhoumai16 plants subjected to 5%, 7.5%, 10%, 12.5%, 15%, 17.5%, and 20% PEG6000 were 0.0128 g, 0.0122 g, 0.0098 g, 0.0090 g, 0.0065 g, 0.0060 g, 0.0052 g, and 0.0026 g, respectively.”
Comment 6: Accepted. As with point 3, this clarification should be added to the Discussion to contextualize the results’ limitations.
Response: We have integrated the relevant clarification into the Discussion section. Thank you very much for your valuable suggestions.
Comment 7a: Nomenclature consistency: The cultivar name alternates between "Zhoumai 16" (with space) and "Zhoumai16" (without space). Uniform formatting must be applied throughout the text.
Response: We have standardized the cultivar name to “Zhoumai16” throughout the manuscript. Many thanks.
Comment 7b: Parental line description: While additional details were added, the text still omits clarification that Zhoumai16 is a winter wheat cultivar.
Response: We have added this information in the Introduction and Materials and Methods sections: “Zhoumai 16 is a winter wheat cultivar widely cultivated in the Huang-Huai wheat region of China.”
Comment 8, 9: Accepted.
Response: Many thanks for your approve.
Comment 10: Figure 1a: Figure 1a: A scale bar must be added to the seedling photograph for reference.
Response: We have added a scale bar to Figure 1a as suggested. Many thanks.
Your suggestions are of paramount importance to us. Following your advice, the quality of our manuscript has significantly improved. We will further supplement our experiments to enhance the article quality, providing more available germplasm, genes, and markers for wheat stress tolerance breeding. Once again, we sincerely thank you for your guidance on the manuscript.
Reviewer 3 Report
Comments and Suggestions for Authors
The authors made an effort and revised and improved the manuscript.
Suggestions that are not compulsory but makes the work look better (since will be published for ever):
- In the abstract I would try to connect better the new sentences with the old text
- In the conclusion I would add what the author gave as a response (adjusted to the text) to one of my queries:
It is important to note that we cannot assign definite probabilities to individual candidate genes at this stage. Pinpointing specific causal genes directly within the highly complex wheat genome is nearly impossible based solely on initial annotation and transcript data. Candidate genes represent our best initial targets derived from these analyses. Our next steps involve integrating fine-mapping, detailed transcriptome analysis, and functional validation studies to ultimately identify and verify the true target gene(s) responsible for the observed phenotype.
The journal can support in connecting some sentences in the manuscript.
Author Response
Response to Reviewer 3
The authors made an effort and revised and improved the manuscript. Suggestions that are not compulsory but makes the work look better (since will be published for ever):
We thank the reviewer for the helpful suggestions to improve the manuscript.
1 In the abstract I would try to connect better the new sentences with the old text.
Response: Accepted. We have revised the abstract to improve coherence and flow between sentences.
2 In the conclusion I would add what the author gave as a response (adjusted to the text) to one of my queries.
Response: We have incorporated the suggested paragraph into the Conclusion section, with slight adjustments to fit the context:
“In this study, the candidate genes were preliminarily screened through bioinformatic annotation and expression profiling analyses. These candidate genes currently serve only as reference targets, as their biological functions remain to be experimentally validated. To systematically characterize these candidate genes, the following research pipeline were applied: Construction of a secondary mapping population coupled with KASP marker development for high-resolution genetic mapping; Comprehensive identification of target genes through integrated transcriptomic and genomic variation analyses; Functional val-idation employing both gene editing (e.g., CRISPR/Cas9) and transgenic complementation approaches. It should be emphasized that the KASP markers utilized in this study were specifically designed as genetic linkage markers rather than functional markers.”
Round 3
Reviewer 2 Report
Comments and Suggestions for Authors
Thank you for your detailed answers.
I strongly recommend that you should include your response 1 to the discussion and (briefly) to the annotation to demonstrate the limitations of the obtained results and to demonstrate your intention to perform control experiments.
Lines 311-320. The issues designation ( e.g. i, ii, iii) should be introduced into the paragraph.
Kind regards,
Reviewer
Author Response
I strongly recommend that you should include your response 1 to the discussion and (briefly) to the annotation to demonstrate the limitations of the obtained results and to demonstrate your intention to perform control experiments. Lines 311-320. The issues designation ( e.g. i, ii, iii) should be introduced into the paragraph.
Response: Accepted. Thank you for your positive feedback and constructive suggestions. We have carefully revised the manuscript according to your recommendations. Specifically, we have incorporated our response regarding the limitation of control experiments into the Discussion section to improve transparency, and have briefly addressed it in the annotation as well. Additionally, the issues in Lines 311–320 have been clearly itemized (i, ii, iii...) to enhance readability and logical structure. We believe these revisions have strengthened the manuscript and are grateful for your guidance throughout the review process.